# PHRF1 promotes the class switch recombination of IgA in CH12F3-2A cells

Jin-Yu Lee[1]☯, Nai-Lin Chou[1]☯, Ya-Ru Yu[2], Hsin-An Shih[1], Hung-Wei Lin[1], Chine-Kuo Lee[2], Mau-Sun Chang (ORCID)[1,3]*

1 Institute of Biochemical Sciences, National Taiwan University, Taipei, Taiwan, 2 Graduate Institute of Immunology, National Taiwan University College of Medicine, Taipei, Taiwan, 3 Institute of Biological Chemistry, Academia Sinica, Taipei, Taiwan

☯ These authors contributed equally to this work.
* mschang@ntu.edu.tw

**Data Availability Statement:** All relevant data are within the manuscript and its Supporting Information files. RNA-seq files are available at URL https://doi.org/10.5061/dryad.cjsxksnb5.

## Abstract

PHRF1 is an E3 ligase that promotes TGF-β signaling by ubiquitinating a homeodomain repressor TG-interacting factor (TGIF). The suppression of PHRF1 activity by PML-RARα facilitates the progression of acute promyelocytic leukemia (APL). PHRF1 also contributes to non-homologous end-joining in response to DNA damage by linking H3K36me3 and NBS1 with DNA repair machinery. However, its role in class switch recombination (CSR) is not well understood. In this study, we report the importance of PHRF1 in IgA switching in CH12F3-2A cells and CD19-Cre mice. Our studies revealed that Crispr-Cas9 mediated PHRF1 knockout and shRNA-silenced CH12F3-2A cells reduced IgA production, as well as decreased the amounts of PARP1, NELF-A, and NELF-D. The introduction of PARP1 could partially restore IgA production in PHRF1 knockout cells. Intriguingly, IgA, as well as IgG1, IgG2a, and IgG3, switchings were not significantly decreased in PHRF1 deficient splenic B lymphocytes isolated from CD19-Cre mice. The levels of PARP1 and NELF-D were not decreased in PHRF1-depleted primary splenic B cells. Overall, our findings suggest that PHRF1 may modulate IgA switching in CH12F3-2A cells.

## Introduction

An effective immune response requests the appropriate subtypes of antibodies. Class switch recombination (CSR) is responsible for changing the heavy chain isotype from IgM to IgG, IgE, or IgA in B lymphocytes. The constant regions of the immunoglobulin heavy chain ($C_H$ genes) are preceded by a 1- to 10-kb repetitive DNA element of switch (S) regions, except the $C_\delta$ region [1–3]. During CSR, the germline transcription from two S regions yields ssDNA substrates for activation-induced cytidine deaminase (AID) to produce high densities of deoxyuracils in both DNA strands. The AID recruitment to the S regions is mediated by SPT5 when RNA polymerase II (RNAPII) is stalled in the S regions, possibly due to the secondary DNA conformations [4–9]. Subsequently, multiple nicks were generated on the non-template and template strands and double strand breaks (DSBs) in the S regions are connected by canonical non-homologous end joining (c-NHEJ) or microhomology-mediated end joining

**Funding:** This work was supported by the Ministry of Science and Technology (MOST 107-2311-B-002-015) to MSC." "NO - The funders had no role in study design, data collection and analysis, decision to publish, or preparation of the manuscript.

**Competing interests:** The authors have declared that no competing interests exist.

(MMEJ), which is mainly dependent on the length of junctional microhomology (MH) sequences. c-NHEJ requires Ku70-Ku80 heterodimer to recognize DSBs and recruitment of DNA-PKcs for downstream signaling. By contrast, MMEJ is initiated by PARP-1 to bind DSBs and then recruits MRN and CtIP for end-resection [10, 11]. This DSB repair converts IgM to other isotypes of immunoglobulins. A couple of factors involved in DNA damage response and double-strand break repair affect CSR *in vivo*, including PARP1/2, MRN, ATM, H2AX, RNF8, RNF168, and 53BP1 [12–21]. As for the evaluation of CSR *in vitro*, murine CH12F3-2A lymphoma cells [22] and primary splenic B cells have been proven to proceed with consistent switching *in vitro*, which could change IgM to other immunoglobulins in response to a variety of stimulations, such as CD40L, IL-4, TGF-β, and LPS.

PHRF1 (PHD and RING finger domain protein 1) is an E3 ligase containing a plant home-odomain (PHD) that binds methylated histones and a RING domain that ubiquitinates sub-strates. The C-terminus of PHRF1 harbors an SRI (Set2 Rpb1 Interacting) domain which is projected to interact with the phosphorylated C-terminal domain (CTD) of Rpb1 [23]. Initial reports regarding PHRF1's function revealed its role in modulating TGF-β signaling in APL development. PHRF1 ubiquitinates TGIF to ensure redistribution of cPML (the cytoplasmic variant of promyelocytic leukemia protein) to the cytoplasm, where Smad2 is phosphorylated in TGF-β signaling. Aberrant PML-RARα fusion protein interferes with PHRF1's binding to TGIF and prevents the TGIF breakdown by PHRF1 [24, 25]. We focused on a distinctive function of PHRF1 in modulating non-homologous end-joining (NHEJ) in which PHRF1 mediates H3K36 trimethylation (H3K36me3) and NBS1, a component of MRE11/RAD50/NBS1 (MRN) complex, to maintain genomic integrity [26]. Our recent data reveals that PHRF1 is associated with the phosphorylated C-terminal repeat domain of Rpb1, the large subunit of RNA polymerase II (RNAPII), through its SRI domain. PHRF1 binds to the proximal region adjacent to the transcription start site of *ZEB1* and promotes the expression of Zeb1 and cell invasion in lung cancer A549 cells [27].

The involvement of PHRF1 in NHEJ and its interaction with Rpb1 prompted us to investigate whether CSR was affected in the absence of PHRF1. To investigate the impact of PHRF1 on CSR, we knocked out the expression of PHRF1 by Crispr-Cas9 editing and measured IgA switching in CH12F3-2A cells. Also, we determined the switching efficacy of immunoglobulins (Igs) in CD19-Cre mice. Interestingly, it turned out that PHRF1 deficiency influenced IgA switching in CH12F3-2A cells but not in mice.

## Materials and methods

### Cell culture

CH12F3-2A cells were maintained in RPMI 1640 supplemented with 10% FBS and 10 mM 2-mercaptoethanol. For the CSR assay, cells were stimulated with 2 μg/ml of anti-mouse CD40 antibody, 10 ng/ml of recombinant murine IL-4, plus 1 ng/ml of recombinant mouse TGF-β1 (CIT) for 72 hr and then analyzed by flow cytometry. Resting B lymphocytes were isolated from 8-week mouse spleens using anti-CD43 Dynabeads (Thermo Fisher Scientific, Waltham, MA) and stimulated to undergo the class switching to IgG1 and IgE with 10 μg/ml of LPS (Sigma-Aldrich, St. Louis, MO) and 10 ng/ml of recombinant mouse IL-4 (R&D Systems, Minneapolis, MN). For switching to IgG3, 10 μg/ml of LPS. For switching to IgA, 5 μg/ml of LPS, 2 ng/ml of TGF-β (R&D Systems), 10 ng/ml of IL-4, 1.5 ng/ml of IL-5 (BD Pharmingen), and 10 ng/ml of anti-IgD dextran (FinaBio, Rockville, MD).

## Immunoblotting

Cell extracts were solubilized in RIPA buffer and immunoblotted by various antibodies. Antibodies used in Fig 3 were listed in S1 Table in S1 File. The anti-PHRF1 monoclonal antibody has been described previously [26].

## sgRNA-mediated PHRF1 deletion

PHRF1 knockout CH12F3-2A cells were carried out by Cas9 RNP nucleofection. Briefly, Cas9 recombinant protein was produced as previously described [28, 29]. Two sgRNA oligos were designed to delete the exon 2 of mPHRF1, which contains the ATG translation start site. The sequences were sgRNA#1, 5'-TAATACGACTCACTATAGGTCATCCATGGCTGCACATG TTTTAGAGCTATGCTGGAAACAGCATAGCAAGTTAAA -3' and sgRNA#2, 5'-TAATAC GACTCACTATAGTGACATTTAAGCTCCCAAGGTTTTAGAGCTATGCTGGAAACAGCATAGCA AGTTAAA-3'. Cas9 RNP complexes were assembled right before nucleofection by mixing equal volumes of 40 μM of Cas9 protein and 48 μM of sgRNAs at a molar ratio of 1:1.2 and incubating at 37˚C for 15 min. Nucleofection reaction consisted of 1 x $10^6$ of CH12F3-2A cells in 20 μl of nucleofection buffer, 2 μl for two sets of Cas9 RNP (equivalent to 40 pmol). The nucleofection mixtures were transferred into a 16-well strip in Lonza 4D Nucleofector for nucleofection. The program was set at the pulse code CA-137. After nucleofection, the cells were transferred to the culture plate with a complete culture medium after nucleofection. Subsequently, a serial dilution to obtain complete knockout clones was conducted. Candidate clones were first screened by PCR and then verified by Western blotting.

## Quantitative Real-Time PCR (RT-qPCR)

Total RNA was prepared using TRIzol reagent (Life Technologies, Waltham, MA) with three duplicates. Reverse transcription would be performed using the 2x one-tube RT mix (Bioman, Taipei, Taiwan). The qPCR analysis was performed using 2 x qPCR master mixes (Bioman, Taipei, Taiwan) on the CFX384 Touch™ following the manufacturer's protocol. The reaction included 5 μl of cDNA (5μg) and 1 μM of indicated primers in a final volume of 20 μl master mix. The following thermal profile was used: an initial 30 s denaturation step at 95˚C, followed by 40 cycles respectively at 95˚C for 15 s, 55˚C for 15 s, and 72˚C for 20 s. The data were analyzed using 7500 software v.2.0.1 (Applied Biosystems, Foster City, CA, USA). Gene expression levels would be normalized with an endogenous control glyceraldehyde-3-phosphate dehydrogenase (GAPDH) mRNA. Primer sequences were listed in S2 Table in S1 File.

## Electroporation

CH12F3-2A cells were diluted to 5 x $10^6$ cells/ml in BTXPRESS electroporation solution (BTX, Cambridge, UK) with 20 μg/ml plasmid DNA. Approximate 250 μl of DNA mixture was transferred to one cuvette (BTX, 2 mm gap) and proceeded electroporation with BTX Gemini SC2 Twin Wave Cuvette Electroporator. The program was set with voltage 260, capacitance 950 μF, resistance 50 ohms. After electroporation, cells were transferred to complete growth medium and incubated in 37˚C incubator for 48 hr.

## Mice

All experimental procedures were carried out by protocol #17-02-1051 approved by Institutional Animal Care and Use Committees at Academia Sinica. PHRF1$^{fl/fl}$ mice have been described previously [30] and were bred with CD19-Cre (The Jackson Laboratory, Bar Harbor, ME). CD19$^{cre/+}$PHRF1$^{fl/+}$ mice were then crossed with PHRF1$^{fl/fl}$ mice to produce CD19$^{cre/}$

+PHRF1$^{fl/fl}$ and control littermates, such as PHRF1$^{fl/fl}$, for experimental uses. Mice were reared in the animal facility under a 12-h light/dark cycle with free access to food and water and maintained under pathogen-free conditions at a 12-h day-night cycle. To alleviate their suffering, mice were housed and minimized the number of animals in appropriate cages. Control and CD19$^{Cre/+}$PHRF1$^{fl/fl}$ mice were sacrificed using carbon dioxide inhalation. The animals were monitored closely during the procedure to ensure that they were not experiencing any pain or discomfort. To relieve their pain or distressing, inhalant anesthetics, such as isoflurane, might be used.

### Flow cytometry

For CSR from IgM to IgA, CH12F3-2A cells were stained with Dylite488-conjugated anti-mouse IgA (Abcam, Cambridge, UK) and APC-conjugated anti-mouse IgM (BD Pharmingen). For CSR from primary B cells, cells were stained with fluorochrome-conjugated anti-IgG1, anti-IgG$_3$, anti-IgE, anti-CD19, anti-B220, anti-CD43, anti-BP1, anti-CD24, anti-CD3, anti-IgD, anti-CD21, and anti-CD23 antibodies (all from BD Pharmingen). Cells were then processed on a BD FACSCanto flow cytometry system and the data were analyzed using FlowJo software (BD Pharmingen).

### CSR join sequences in the S region

CSR joins were amplified by nested PCR using genomic DNA prepared from CH12F3-2A cells stimulated with CIT to undergo CSR for 4 days. Sμ primers for Sμ–Sα joins were: SμOut, 5′-AAG TTG AGG ATT CAG CCG AAA CTG-3′; and SμIn, 5′-GGA GAC CAA TAA TCA GAG GGA AGA-3′. Sα primers for Sμ-Sα joins were: SαOut, 5′-AGC ACT GAG TTT AAC AAT CCA GC-3′; and SαIn, 5′-CCT CAG TGC AAC TCT ATC TAG GT-3′. PCR products were cloned into TA vectors, and DNA sequences were analyzed using EMBOSS Needle Pairwise Sequence Alignment Tool.

### Statistical analysis

Analysis was carried out using GraphPad Prism 6 software. All values were expressed as mean ± SD. The paired Student's $t$-test (two-tailed) was used to calculate the statistical significance of differences between groups. The $p < 0.05$ was considered statistically significant.

## Results

### PHRF1 ablation reduced IgA CSR in CH12F3-2A cells

To decipher the impact of PHRF1 deficiency on CSR, we knocked out *PHRF1* gene by deleting the exon 2 containing the ATG codon (a.a. 1–32) using CRISPR-Cas9 gene editing in CH12F3-2A cells. Genotyping results revealed that exon 2 of the *PHRF1* gene was deleted in two independent clones (KO#1 and KO#2) (Fig 1A). KO#1 and KO#2 clones did not have the expression of PHRF1 through immunoblotting analysis (Fig 1B). To investigate the effect of PHRF1 on CSR, control and PHRF1 KO cells were stimulated with CD40L, IL-4, and TGF-β (CIT) for three days to induce IgA switching, and then IgA level was measured using flow cytometry. As expected, the proportion of IgM switching to IgA was remarkably reduced in PHRF1-depleted CH12F3-2A cells compared with control cells (Fig 1C). When full-length PHRF1 was reintroduced into *PHRF1* KO cells, IgA production was restored (Fig 1D), further confirming that PHRF1 was essential for IgA switching in CH12F3-2A cells.

To rule out the off-target effects of CRISPR-Cas9 editing as the cause of the observed phenotype, an element of shRNA targeting a.a. 95–102 to silence PHRF1's expression was

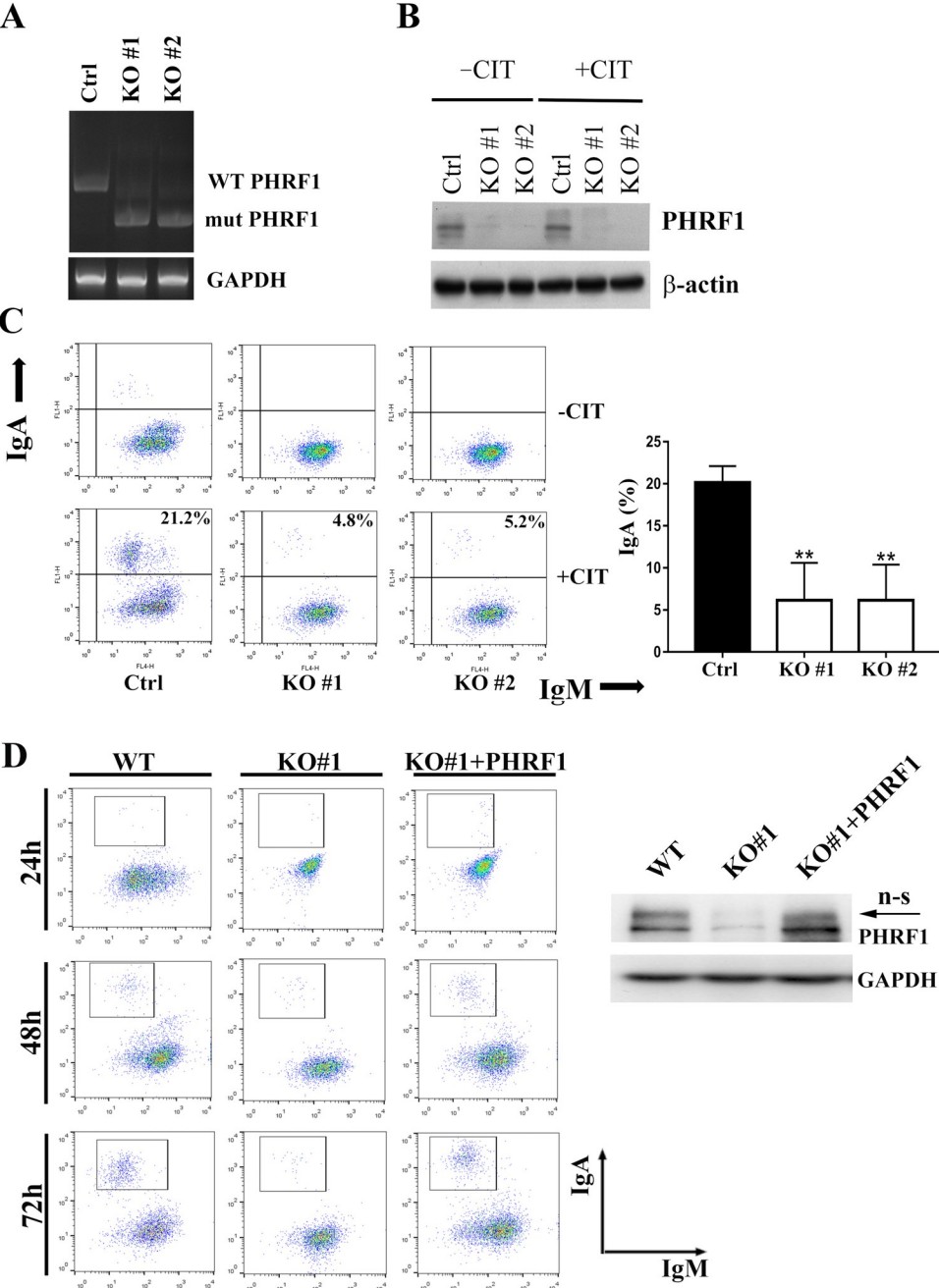

**Fig 1. PHRF1 knockout reduced IgA switching.** (A) CH12F3-2A cells were nucleofected with two independent sgRNA oligos to target exon 2 of the *PHRF1* gene. Positive clones were selected by serial dilution. Genotyping PCR was conducted using 5'- and 3'-primers. GAPDH primers were used as a control. (B) Cell extracts harvested from CH12F3-2A (WT) and PHRF1-depleted cells (KO#1 and KO#2) with the treatment of anti-CD40 antibody, IL-4, and TGF-β (CIT) for 3 days were immunoblotted with indicated antibodies. All Western blots were processed in identical conditions and cropped from S9 Fig in S1 File. (C) WT and PHRF1 depleted cells were either untreated or stimulated with CIT for 3 days. IgA population was analyzed by flow cytometry. The percentages of IgA/IgM from three independent experiments were presented as mean ± SD. *** P < 0.001 by Student's *t*-test. (D) PHRF1 KO#1 cells were transduced with Lenti-PHRF1 and selected with blasticidin. Cell extracts were harvested and immunoblotted with indicated antibodies. n.s., non-specific. All Western blots were processed in identical conditions and cropped from S9 Fig in S1 File. Representative images of IgA switching at 24 h, 48 h, and 72 h post-CIT treatment were determined by flow cytometry.

conducted (S1A Fig in S1 File). We found a similar reduction in IgM shifting to IgA as seen in PHRF1 KO cells (S1B Fig in S1 File).

## PHRF1 deficiency did not affect cell proliferation and germline transcription

To avoid the possibility that IgA reduction was caused by defective cell proliferation, we measured the cell density using carboxyfluorescein diacetate succinimidyl diester (CFSE) dilution assay. Control and PHRF1 deficient cells showed a similar CFSE intensity post 6, 24, and 48 h labeling (S2A Fig in S1 File). Additionally, PHRF1-depleted CH12F3-2A cells were left untreated or treated with LPS, LPS+IL4, or anti-CD40 Ab+IL4 for four days, and then total cell numbers were calculated. The result showed similar cell expansion in control and PHRF1 deficient CH12F3-2A cells upon different stimulations (S2B Fig in S1 File). Additionally, the time course of CSR at 24 h, 48h and 72 h post CIT stimulation revealed that *PHRF1* KO cells could not undergo CSR in each round of cell division (S2C Fig in S1 File), further supporting the importance of PHRF1 in CSR progression.

Germline transcripts (GLTs), initiating from the upstream I promoters (I) and proceeding through the switch region and $C_H$ exons, facilitate the cytosine deamination by AID that is required for CSR [3]. To measure the expression level of GLTs, a quantitative RT-PCR (qRT-PCR) analysis was conducted. Comparable results of Iμ and Iα GLTs either untreated or post-CIT stimulation were found in control and PHRF1 depleted CH12F3-2A (S3 Fig in S1 File), indicating that inactivation of PHRF1 did not affect the expression of germline transcripts to reduce IgA production.

## PHRF1 deficiency did not significantly affect the microhomology of Sμ-Sα junction

Microhomology (MH) could be used as a bridge to align the breaking ends, in which longer MHs (2–20 nucleotides) are favorable for MMEJ [10, 11, 31]. To address the repair choice upon PHRF1 deficiency, we analyzed MH of Sμ-Sα joining in control and PHRF1 depleted CH12F3-2A cells post CIT stimulation. We conducted the nested PCR to amplify the joining junctions of Sμ and Sα. Each of the clones containing junction fragments from control (n = 41) and PHRF1 (n = 39) depleted cells was sequenced. The alignment of MHs was listed in S4 Fig in S1 File. Analyses of CSR junctions revealed that most of the junction sequences in control cells had a mean overlap length 2.6 bp at the junction. Similarly, *PHRF1*-deficient clones had a the mean overlap length 2.7 bp at the junction (Fig 2A). This suggests that IgA switching might not utilize a longer MH in the absence of PHRF1. MH levels ≥ 5–10 bps were similar in control and PHRF1-depleted cells (Fig 2B), indicating that other end-joining mechanisms might be involved in the absence of PHRF1.

## PHRF1 depletion reduced the expression levels of PARP1 and NELFs

We examined the protein levels of factors participating in transcriptional regulation or DNA damage repair by immunoblotting analysis. Reduced IgA switching might not be the consequence of the aberrant TGF-β signaling, since phospho-Smad2 on S465/S467 was unchanged (Fig 3A, right panel). Most of the DNA damage-related factors, except PARP1, and the level of γ-H2AX were also unchanged (Fig 3A, right panel), indicating that DNA damage response or repair proteins were not affected in PHRF1-depleted cells. Instead, the expression levels of NELF-A, NELF-D, and H3K36me2/me3 were reduced in PHRF1 KO cells (Fig 3A, left panel). Quantitative results from three independent experiments confirmed that PARP1, NELF-D, and

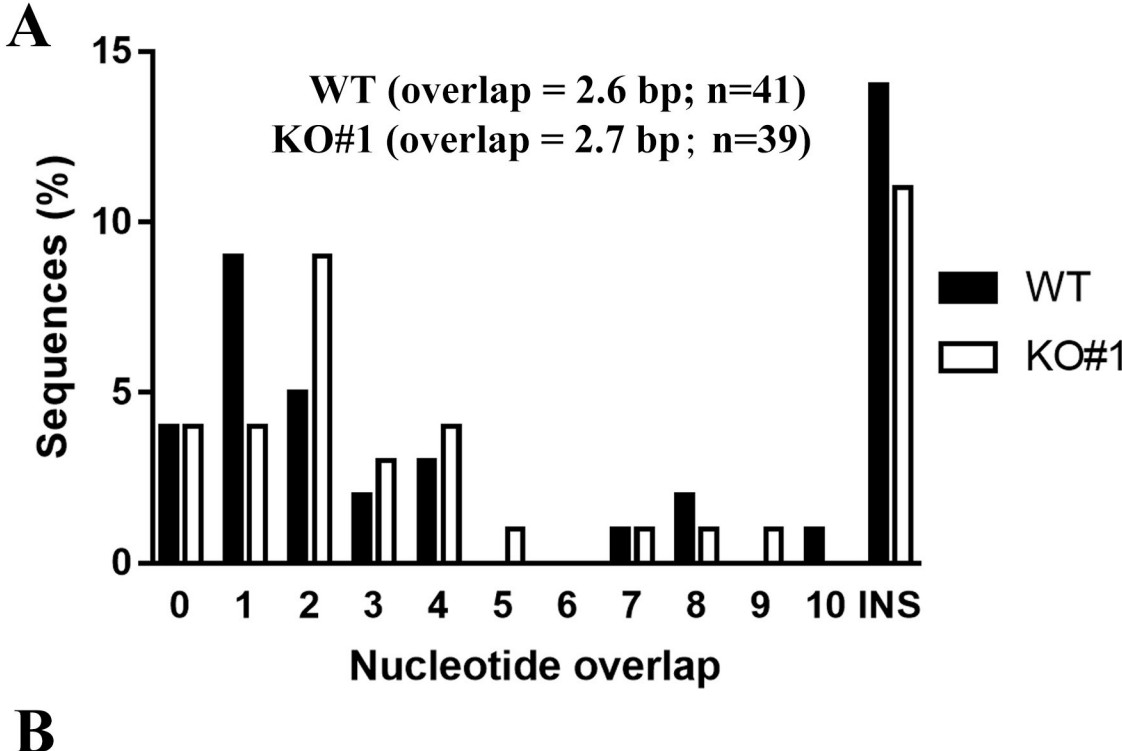

**Fig 2. Analysis of microhomology in Sμ-Sα joining.** (A) The percentage of switch junctions with different lengths of microhomology was shown with the percentage of nucleotide overlap. Sequence data were compiled from three independent experiments. (B) The spectrum of junctions in control versus PHRF1 KO#1 cells was tabulated. The difference in the percentage of junctions with microhomology of larger than 2 nucleotides or more was not statistically significant.

H3K36me3 were significantly decreased in PHRF1 KO cells (Fig 3B). Additionally, the phosphorylation status of Rpb1's CTD represents the different phases of initiation, pausing, elongation, and termination in RNA transcription. Phosphorylation on S5, a marker for transcription initiation, remained unchanged. By contrast, the phosphorylation level of Rpb1's CTD on S2, a signature at the transcription elongation, was reduced in PHRF1-depleted CH12F3-2A cells (Fig 3C), indicating that the absence of PHRF1 might alter the phosphorylation signature on the CTD region to stall Rpb1 at transcription elongation. We introduced PARP1 and NELF-D into PHRF1 KO#1 cells by electroporation and confirmed their expressions by immunoblotting. Flow cytometry result showed that PARP1 could significantly elevate IgA production, while NELF-D was unable to increase IgA production in KO#1 cells, strongly suggesting that PARP1 might be the main downstream target by PHRF1 depletion (Fig 3D).

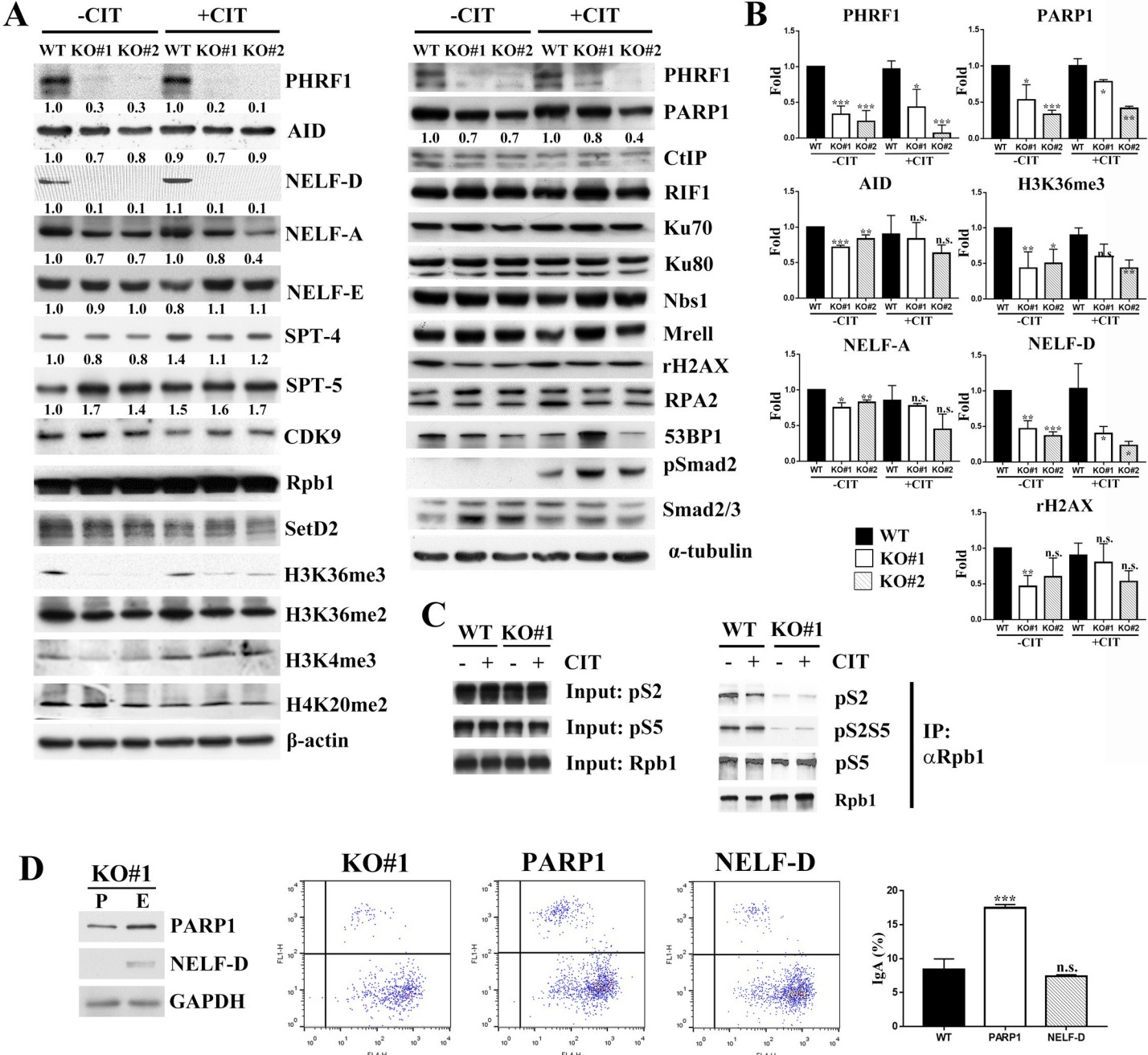

**Fig 3. PHRF1 depletion reduced the amounts of PARP1, NELF-A, and NELF-D.** (A) WT and PHRF1 depleted CH12F3-2A cells were either untreated or stimulated with CIT for 3 days. Cell extracts were harvested and immunoblotted with indicated antibodies. Note that PARP1, NELF-A, NELF-D, Spt4 were reduced in PHRF1 deficient cells. (B) WT and PHRF1 KO#1 CH12F3-2A cells were stimulated with CIT for 3 days. Cell extracts were isolated, subjected to immunoblotted analysis, and normalized to α-tubulin. Three independent experiments were presented as mean ± SD. * P <0.05, ** P < 0.01, *** P < 0.001 by Student's *t*-test. n.s., not significant. (C) WT and PHRF1 KO#1 CH12F3-2A cells were either untreated or stimulated with CIT for 3 days. Cell extracts were immunoprecipitated with anti-Rpb1 antibody and immunoblotted with anti-phospho-CTD antibodies. All Western blots were processed in identical conditions and cropped from S8 Fig in S1 File. (D) PHRF1 KO#1 cells were electroporated with PARP1 or NELF-D plasmid. Cell extracts were harvested and immunoblotted with indicated antibodies. P: parental KO#1 cells. E: electroporated KO#1 cells. All Western blots were processed in identical conditions and cropped from S9 Fig in S1 File. Representative images of IgA switching at 72 h post-CIT treatment were determined by flow cytometry.

To gain more information regarding the global landscape of gene expression in the absence of PHRF1, RNA-seq analysis was conducted. Approximately 750 differentially expressed genes (DEGs) were obtained, in which fold change > 2 (up-regulated) or < 0.5 (down-regulated) and PPEE < 0.05 were considered as statistically significant. Subsequently, these DEGs were subjected to the Gene Ontology (GO) analysis. The result showed that several distinctive GO categories could be grouped by correlated DEGs, including the positive regulation of RNA polymerase II (Fig 4A). Among these DEGs, top 50 up-regulated and down-regulated genes were clustered for heat map analysis (Fig 4B) and factors involved in the positive regulation of RNAPII transcription were also subjected for heat map analysis (Fig 4C). To confirm RNA-seq results, RT-qPCR was carried out. The mRNA levels of genes involved in the positive RNAPII regulation, such as Lef1, Trp73, Trp53inp1 in Fig 4C, and PARP1, NELF-A, NELF-D in Fig 3A, were genuinely decreased by RT-qPCR analyses (Fig 4D). We also measured the mRNA levels of SetD2, Amyd2, and Amyd5, which are responsible for the methylation of H36K36me2/3. However, the mRNA levels of these histone methyltransferases were not changed in RT-qPCR analysis (Fig 4D). Furthermore, to address whether PHRF1 deficiency affected TGF-β signaling, we clustered the entire components or FC > 2 in TGF-β signaling (KEGG#04350). These two heatmaps were shown in S5 Fig in S1 File and the P-value of TGF-β signaling was 0.56 in the pathway enrichment analysis, suggesting that PHRF1 KO might not affect TGF-β signaling.

## Generation of PHRF1 knockout in CD19-Cre mice

To evaluate the impact of PHRF1 deficiency on CSR *in vivo*, we inactivated the expression of PHRF1 by crossing PHRF1[fl/fl] mice with CD19-Cre transgenic mice to disrupt PHRF1's expression in B lymphocytes. PHRF1[fl/fl] mice harboring two loxP elements to flank the exon 2 to 9 (a.a. 1–343) of the murine *PHRF1* gene were described previously (S6 Fig in S1 File) [30]. We took advantage of CD19-driven Cre recombinase specifically expressed in B cell progenitors [32] to knockout PHRF1's expression in the B lymphocytes. As a result, two functional domains, E3 Ring domain and PHD domain (a.a. 109–153 and a.a. 188–232, respectively), were deleted by Cre recombinase in the lineage of B cells. Cd19[Cre/+]PHRF1[fl/fl] pulps were viable without noticeably developmental defects.

To assess whether PHRF1 deficiency affected B cell development, we examined the distribution of B cell subsets isolated from the bone marrow and spleen using different surface markers of lymphocytes. B cell lineage, including Pre-pro-B cells, Pro-B and Pre-B, could be distinguished based on their differential expression of CD43, B220, CD24, and BP1 in the bone marrow. Immature and mature B cells could be segregated based on the differential expression of IgM and IgD. The result showed that control and Cd19[Cre/+]PHRF1[fl/fl] B cells exhibited a similar proportion of surface markers in the bone marrow and spleen (S7 Fig in S1 File), indicating that PHRF1 ablation did not interfere with the production of B cells in Cd19[Cre/+]PHRF1[fl/fl] mice.

## PHRF1 deficiency did not affect the CSR of immunoglobulins in CD19-Cre mice

To determine any CSR defect in PHRF1-deficient B cells, resting B cells were isolated from the spleen using anti-CD43 microbeads and IgM was induced to switch to other Ig isotypes *ex vivo* and then determined by flow cytometry. Unexpectedly, CSR to surface IgG1, IgG2a, IgG3, and IgA was statistically unchanged in the splenic B cells of control and Cd19[Cre/+]PHRF1[fl/fl] mice (Fig 5A). We harvested splenic B cell extracts from 7-week littermates for the immunoblot analysis. Contrary to CH12F3-2A cells, the expression levels of PARP1, NELF-A, NELF-D, and H3K36me2/me3 were not remarkably decreased in primary B cells derived from Cd19[Cre/

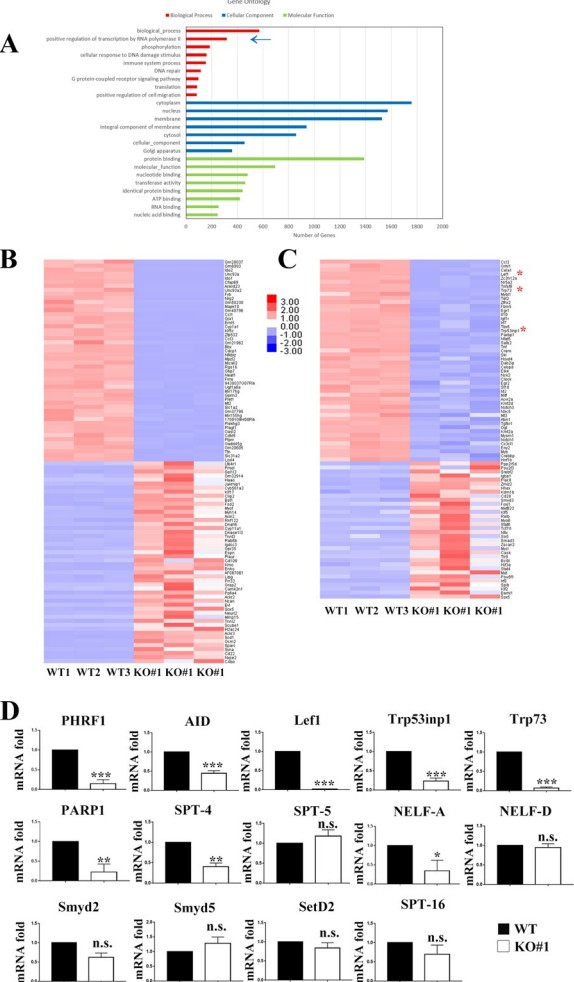

**Fig 4. RNA-seq analysis in WT and PHRF1 KO#1 cells.** (A) Identification of molecular function, cellular component, and the biological process by Gene Ontology software based on the differentially expressed genes (DEGs) in WT and PHRF1 KO#1 cells. Note that a blue arrow marked the positive regulation of transcription by RNAPII. (B) The heatmap was generated by clustering the top 50 up- and down-regulated genes (n = 3 per group). Down-regulated genes are shown in blue shades and up-regulated genes are shown in red shades. (C) The heatmap by clustering the positive regulation of transcription by RNAPII (n = 3 per group). (D) WT and PHRF1 KO#1 CH12F3-2A cells were stimulated with CIT for 3 days. Total RNAs were isolated and subjected to RT-qPCR and normalized to the expression level of GAPDH. Three independent experiments were presented as mean ± SD. * P <0.05, ** P < 0.01, *** P < 0.001 by Student's *t*-test. n.s., not significant.

[+] PHRF1[fl/fl] mice (Fig 5B). We determined the cell numbers in control and PHRF1 deficient B lymphocytes. Similar numbers of nucleated cells (2.1 x 10$^7$ versus 2.0 x 10$^7$ cells/ml) were found in control and Cd19[Cre/+]PHRF1[fl/fl] mice (Fig 5C). To monitor cell proliferation in cultured splenic B cells, CFSE labeling was conducted to quantify cell division. After three days in culture, most of the cells in all stimulated B cells exhibited no comparable difference between control and Cd19[Cre/+]PHRF1[fl/fl] B cells (Fig 5D). Furthermore, by stimulating primary B cells in varying concentrations of TGF-β (1, 2, and 5 μg/ml), we measured the proliferation level using CFSE staining (S8A Fig in S1 File) and CSR to IgA in primary B cells (S8B Fig in S1 File). The results showed that PHRF1-KO B cells were not responsive to TGF-β stimulation. Finally, Iμ and Iα GLTs for IgA induction were also comparable in control and

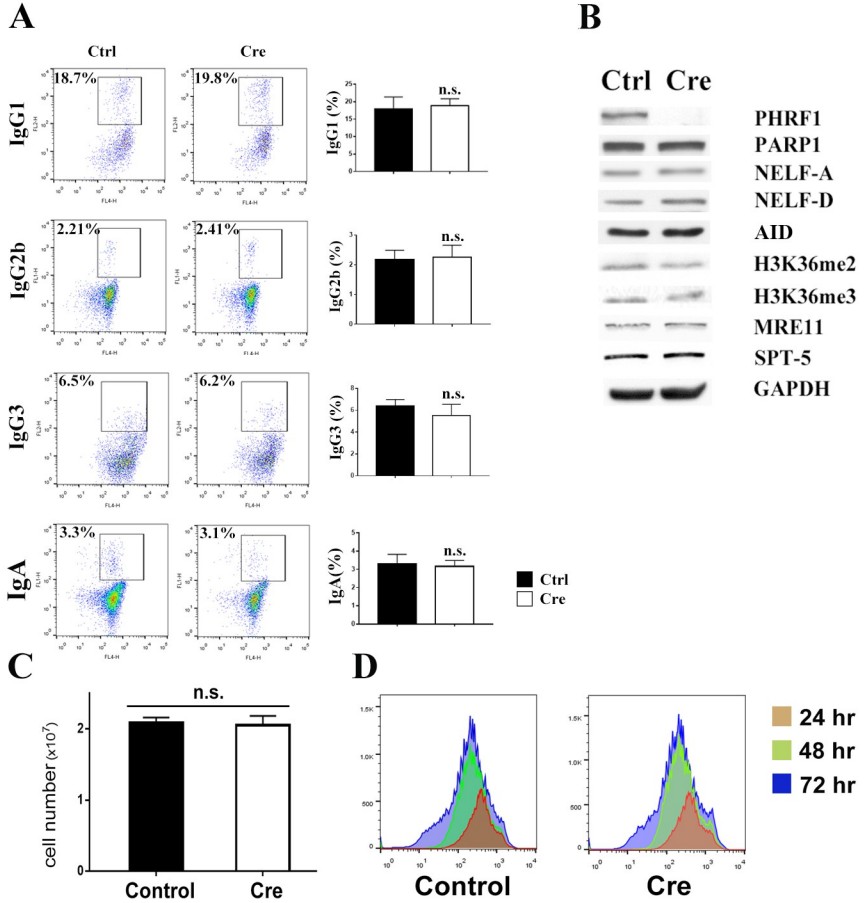

**Fig 5. PHRF1 deficiency did not impair CSR in primary splenic B cells.** (A) Splenic B cells from control and Cd19$^{Cre/+}$ PHRF1$^{f/f}$ mice were stimulated to proceed CSR for 72 h. Representative images of IgG1, IgG2a, IgG3, and IgA were determined by flow cytometry. Quantitative results were measured by three independent experiments shown in the right panels. n.s., not significant by Student's *t*-test. (B) Cell extracts harvested from splenic B cells were subjected to immunoblotting analysis with indicated antibodies. All Western blots were processed in identical conditions and cropped from S9 Fig in S1 File. (C) Control and Cd19$^{Cre/+}$ PHRF1$^{f/f}$ B cells were stimulated for IgA switching and cell numbers were determined after four days. Quantitative results for three independent experiments. n.s., not significant by Student's *t*-test. (D) B cell proliferation was monitored by labeling carboxyfluorescein succinimidyl ester (CFSE).

PHRF1-depleted primary B cells (S8C Fig in S1 File). Taken together, PHRF1 deficiency might not be able to affect the CSR of immunoglobulins in primary splenic B cells.

## Discussion

In light of impaired NHEJ in U2OS and HEK293 cells due to PHRF1 silencing [26], we intended to explore whether PHRF1 ablation could affect CSR *in vivo*. As expected, the inactivation of PHRF1 reduced IgA switching and the expression of PARP1, NELF-A, and NELF-D was remarkably reduced in CH12F3-2A cells, leading to our assumption that PHRF1 depletion might alter the CSR process in mice. Although we carried out five different methods to induce CSR in primary B cells [33–37], PHRF1 ablation was not able to affect the switching of IgM to IgG1, IgG2a, IgG3, and IgA in CD19-Cre mice. This suggests that there may be other compensatory mechanisms in place *in vivo* that are able to offset the loss of PHRF1. The molecular basis for these compensatory mechanisms is not yet clear and may be an interesting area in animals.

Stalled RNAPII at the switch region facilitates the AID targeting to Ig locus. A recent study suggest that AID is recruited to the S regions by Spt5 when RNAPII is stalled [4]. Several components of RNAPII "stalling" machinery, including those associated with the C-terminal repeat domain (CTD) of Rpb1, play critical roles in generating diversified antibodies during CSR [38–40]. RNAPII pauses after transcribing 20–40 nucleotides due to DRB sensitivity-inducing factors (DSIF, Spt4 and Spt5) and negative elongation factors (NELF-A to -E). Subsequently, positive transcription elongation factor b (P-TEFb) phosphorylates CTD of Rpb1 on S2 and CDK9 phosphorylates DSIF, leading to the dissociation of NELFs from RNAPII and transcription elongation to proceed [41]. Although our results were not sufficient to conclude that AID targeting was directly affected by the absence of PHRF1, the reduced phosphorylation of Rpb1 on S2 in the CTD and decreased levels of NELF-A and NELF-D might indirectly disturb the interaction between AID and Spt5, leading to the interference in AID targeting to the switch region. Therefore, the absence of PHRF1 may result in aberrant transcription progress and a defective CSR for IgA switching.

PARP-1 is involved in MMEJ for binding to DSBs by competing with Ku70-Ku80 and facilitating the recruitment of end-resection factors [10]. However, the precise mechanism by which PHRF1 deficiency affects the efficiency of CSR is not fully understood. Unlike DSBs repaired by c-NHEJ, DSBs repaired by MMEJ need more MHs between the donor and acceptor switch regions. c-NHEJ generally utilizes 0–2 bp MHs; by contrast, MMEJ tends to use more than 2–20 bp MHs. Our data showed a lower expression level of PARP1 in CH12F3-2A cells; therefore, PHRF1 absence may compromise MMEJ efficiency for IgA CSR. Although the switch junctions in PHRF1 deficiency were not biased toward a longer microhomology, other end-joining mechanisms might be active in the absence of PHRF1.

In summary, the present data not only corresponded to our previous report that PHRF1 silencing affects NHEJ, but indicated the fact that the absence of PHRF1 impairs CSR, at least in CH12F3-2A cells. We also found that PARP1 was important for IgA production in PHRF1-depleted CH12F3-2A cells. Additionally, decreased NELF-A and NELF-D may affect the transcription elongation favorable for IgA switching. While the exact molecular mechanism is not clear; we provide evidence to show that PHRF1 does play a role of CSR in CH12F3-2A cells.

## Supporting information

**S1 File.**
(DOCX)

## Acknowledgments

We are grateful to Dr. Sheng-Wei Lin (IBC, Academia Sinica) for the technical assistance of flow cytometry and fluorescent microscope. We thank Adaobi Amanna for English grammar and spelling editing.

## Author Contributions

**Conceptualization:** Mau-Sun Chang.

**Data curation:** Jin-Yu Lee, Nai-Lin Chou, Ya-Ru Yu, Hsin-An Shih, Hung-Wei Lin, Chine-Kuo Lee.

**Formal analysis:** Jin-Yu Lee, Nai-Lin Chou, Ya-Ru Yu, Hung-Wei Lin.

**Funding acquisition:** Mau-Sun Chang.

**Investigation:** Chine-Kuo Lee.

**Methodology:** Jin-Yu Lee, Nai-Lin Chou, Ya-Ru Yu, Hsin-An Shih, Hung-Wei Lin.

**Resources:** Hsin-An Shih.

**Software:** Jin-Yu Lee, Nai-Lin Chou, Hung-Wei Lin.

**Writing – original draft:** Mau-Sun Chang.

**Writing – review & editing:** Mau-Sun Chang.

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
