## [Decision Letter · Decision Letter 0]

8 Feb 2023

PONE-D-23-01559PHRF1 promotes the class switch recombination of IgA in CH12F3-2A cellsPLOS ONE

Dear Dr. Chang,

Thank you for submitting your manuscript to PLOS ONE. After careful consideration, we feel that it has merit but does not fully meet PLOS ONE’s publication criteria as it currently stands. Therefore, we invite you to submit a revised version of the manuscript that addresses the points raised during the review process.

We look forward to receiving your revised manuscript.

Kind regards,

Yan Shan, PhD

Academic Editor

PLOS ONE

“We are grateful to Dr. Sheng-Wei Lin (IBC, Academia Sinica) for the technical assistance of flow cytometry and fluorescent microscope. We thank the National C6 RNAi Core Facility at Academia Sinica in Taiwan for providing CRISPR/Cas reagents and related services. This work was supported by the Ministry of Science and Technology (MOST 107-2311-B-002-015) to MSC.”

“NO - Include this sentence at the end of your statement: The funders had no role in study design, data collection and analysis, decision to publish, or preparation of the manuscript.”

8. In your Data Availability statement, you have not specified where the minimal data set underlying the results described in your manuscript can be found. PLOS defines a study's minimal data set as the underlying data used to reach the conclusions drawn in the manuscript and any additional data required to replicate the reported study findings in their entirety. All PLOS journals require that the minimal data set be made fully available. For more information about our data policy, please see http://journals.plos.org/plosone/s/data-availability.

Additional Editor Comments:

Thank you for submitting your manuscript to PLOS ONE. Your manuscript has been peer reviewed by two referees (attached below), and some major/minor concerns have been raised. Although the current format is not acceptable for publication, we have decided to invite you to revise your manuscript (Major revision) after careful consideration. If you decide to resubmit, please provide a point-by-point response to the concerns raised, and highlight the changes in the revised manuscript. We look forward to your revision.

Reviewers' comments:

Reviewer's Responses to Questions

**Comments to the Author**

1. Is the manuscript technically sound, and do the data support the conclusions?

Reviewer #1: Partly

Reviewer #2: Partly

2. Has the statistical analysis been performed appropriately and rigorously? 

Reviewer #1: Yes

Reviewer #2: No

3. Have the authors made all data underlying the findings in their manuscript fully available?

Reviewer #1: Yes

Reviewer #2: No

4. Is the manuscript presented in an intelligible fashion and written in standard English?

Reviewer #1: Yes

Reviewer #2: No

5. Review Comments to the Author

Reviewer #1: In this manuscript, Lee et al. reported that IgA production, PARP1, NELF-A and NELF-D was decreased in PHRF1 KO and knockdown CH12F3-2A cells. Further, they found that gG1, IgG2b, IgG3 and IgA switching were not significantly affected in PHRF1 deficient splenic B lymphocytes isolated from CD19-Cre mice. Though this study addresses a novel area related to IgA switching and PHRF1, but the authors need to clarify their model, statistical analysis and experimental design. Moreover, Further experiments are required and rationale is needed to delineate the model.

Below concerns (major and minor) should be address in a revised manuscript before acceptance for publication.

Major concerns:

1. The quality or resolution of all of figures.

The resolution of all these figures is too low to get effective and detailed information. I can’t get any useful information from Figure 2A. The titles of vertical axis on Figure 3B were very difficult to distinguish. The percentage of cells labeled on top-left corner on Figure 4A were also very fuzzy. So, please provide high resolution figures on revised version.

2. Although RNA-seq results were still needed to be further investigated, the validation of this raw transcriptomic data is required for publication to ensure these data were solid and repeatable. Typically, the validation method is that more than 10 genes’ expression in RNA-seq should be validated by RT-qPCR.

3. Quantification of blots on Figure 3A.

The western blot results on Figure 3A are need to repeat and statistical analysis for quantification, especially for PARP1, γ-H2AX, NELF-A, NELF-D, and H3K36me2/me3 (significantly changed proteins).

4. The experimental design for Figure 3C was inappropriate. In “Input” part, pS2, pS2S5 and pS5 should be included to indicate these proteins in WT and KO cell extracts were equal. Rpb1’s blot should also be included in IP results as a control.

5. The innovation of this study.

As mentioned by the authors in end of discussion, the molecular explanation of this study is too “obscure”. In current study, the authors performed lots of molecular experiments, but no reasonable mechanism raised. A molecular explanation is required, at least can partially explain these phonemes.

A suggestion may enhance this study’s innovation mentioned below:

Base on the results from CH12F3-2A cells and CD19-Cre mice, PARP1, NELF-A and NELF-D seem to play important roles in PHRF1 regulating CSR process. I do not insist on the following hypothesis but as a possible illustration of how the current results might be interpreted: PARP1, NELF-A or/and NELF-D maybe the key down-stream effectors in PHRF1 regulating CSR progression pathway. Obviously, we need more experiments (express or co-express PARP1, NELF-A and NELF-D in PHRF1 KO cells, then test whether it can rescue or partially rescue the CSR defect) to support the hypothesis. The hypothesis I mentioned is just a possibility and it is more than welcome if you have other regulation mechanism which can be supported by your data. A molecular explanation raised and confirmed by your work would be a shining point in your manuscript.

Minor concerns:

6. In Figure S1 panel A, there were two bands on PHRF1’s blot. Please indicate which band is PHRF1 and which is unspecific band.

7. The complementary experiment in Figure S1 was a very solid evidence to confirm that PHRF1 KO directly compromised CSR progression. Tt is highly recommended to move these results to Figure 1.

8. The abbreviation of “CIT” should be described in the legend of Figure 1B rather than Figure 1C.

9. Lack of Data Availability statement.

Reviewer #2: The manuscript by Lee et al, describes the effects of PHRF1 deletion on class-switching in B cells. Somewhat surprisingly, PHRF1 deletion led to a substantial decrease in IgA switching in CH12F3 cells but not in primary B cells. Though the authors start with the premise that PHRF1 is a regulator of TGFB signaling, they instead focused on the differential regulation of transcription elongation in PHRF1 KO cells. There are several major issues with the manuscript in its present form. I would advise a major revision before the manuscript is considered for publication. My comments are listed below:

1. The logic behind investigating transcriptional elongation in PHRF1 KO cells is not clearly defined. Even though, the PHRF1-deficient CH12 cells showed decreased phosphorylation of Rbp1 subunit of RNA polII at S2 position, the germline transcription at IgH locus in CH12 cells was unaffected. In addition, the effects of PHRF1 deficiency on NELF expression and Rbp1 phosphorylation was totally missing in activated primary B cells. The authors should also examine IgH germline transcription in PHRF1 KO primary B cells.

2. Consistent with the role PHRF1 in regulating TGFB signaling, there appears to be a slight increase in the levels of phospho-SMAD2/3 in PHRF1 KO CH12 cells, though this is not quantified. Can the authors test the possibility that the PHRF1-deficent CH12 and primary B cells are differentially sensitive to TGFB stimulation. One can do this by stimulating B cells in varying concentrations of TGFB and measuring the survival, proliferation and CSR to IgA in CH12 and primary B cells.

3. The authors should also provide a more thorough analysis of the transcriptomics data, especially with regards to the transcriptional factors that were identified as being differentially expressed. Perhaps a more careful examination of TGFB signaling could provide readers some more context.

4. The authors observed a slight but non-significant change in the microhomology usage. At what time point were these experiments performed. The repair of switch junctions through MMEJ follows much slower kinetics (PMID: 31806351) and therefore it is better to analyze the microhomology usage at 72 hours post-CIT stimulation. More clones should be screened to calculate the statistical significance of these findings.

5. The authors should refine the writing at several places in the manuscript.

6. The statistical methods used in the study are not detailed enough. The authors should clearly describe the statistical methods used with each figure legend.

6. PLOS authors have the option to publish the peer review history of their article (what does this mean?). If published, this will include your full peer review and any attached files.

Reviewer #1: No

Reviewer #2: **Yes: **Vipul Shukla

---

## [Author Response · Author response to Decision Letter 0]

23 Mar 2023

Reviewer #1: 

Major concerns:

Q1. The quality or resolution of all of figures.

The resolution of all these figures is too low to get effective and detailed information. I can’t get any useful information from Figure 2A. The titles of vertical axis on Figure 3B were very difficult to distinguish. The percentage of cells labeled on top-left corner on Figure 4A were also very fuzzy. So, please provide high resolution figures on revised version.

A1. Thanks for the reminder. We have improved the resolutions of all figures and reorganized Fig 2. The original Fig 2A has been replaced with S4 Fig and fonts for the percentage of cells in Fig 4A has been enlarged. 

Q2. Although RNA-seq results were still needed to be further investigated, the validation of this raw transcriptomic data is required for publication to ensure these data were solid and repeatable. Typically, the validation method is that more than 10 genes’ expression in RNA-seq should be validated by RT-qPCR.

A2. Thanks for the suggestion. We have completed 14 genes’ expression using RT-qPCR and the result was shown in Fig 4B. Most of these genes were significantly down-regulated, which was in line with the RNA-seq results. 

Q3. Quantification of blots on Figure 3A. The western blot results on Figure 3A are need to repeat and statistical analysis for quantification, especially for PARP1, γ-H2AX, NELF-A, NELF-D, and H3K36me2/me3 (significantly changed proteins).

A3. Thanks for the suggestion. Three independent immunoblots were conducted. Quantification of immunoblots were shown in Fig 3B. The protein levels of PARP1, NELF-D, and H3K36me3 were significantly changed, but NELF-D and γ-H2AX were not statistically affected.

Q4. The experimental design for Figure 3C was inappropriate. In “Input” part, pS2, pS2S5 and pS5 should be included to indicate these proteins in WT and KO cell extracts were equal. Rpb1’s blot should also be included in IP results as a control.

A4. Thanks for the suggestion. We have added the inputs of pS2 and pS5 of Rpb1 in Fig 3C. Rpb1’s blot was also included in IP result. 

Q5. The innovation of this study. As mentioned by the authors in end of discussion, the molecular explanation of this study is too “obscure”. In current study, the authors performed lots of molecular experiments, but no reasonable mechanism raised. A molecular explanation is required, at least can partially explain these phonemes. A suggestion may enhance this study’s innovation mentioned below: 

Base on the results from CH12F3-2A cells and CD19-Cre mice, PARP1, NELF-A and NELF-D seem to play important roles in PHRF1 regulating CSR process. I do not insist on the following hypothesis but as a possible illustration of how the current results might be interpreted: PARP1, NELF-A or/and NELF-D maybe the key down-stream effectors in PHRF1 regulating CSR progression pathway. Obviously, we need more experiments (express or co-express PARP1, NELF-A and NELF-D in PHRF1 KO cells, then test whether it can rescue or partially rescue the CSR defect) to support the hypothesis. The hypothesis I mentioned is just a possibility and it is more than welcome if you have other regulation mechanism which can be supported by your data. A molecular explanation raised and confirmed by your work would be a shining point in your manuscript.

A5. Many Thanks for this suggestion. To highlight the importance of PARP1 and NELF-D in PHRF1 KO cells, we have purchased PARP1 and NELF-D plasmids from Sino Biological. (NELF-A was not available.) PARP1 or NELF-D plasmid was electroporated into PHRF1 KO cells and then IgA switching was determined. The result showed that PARP1 significantly increased IgA production, but NELF-D did not (in Fig 3D). This result indicated that PARP1 might be the main target due to PHRF1 deficiency in PHRF1 KO cells. 

Minor concerns:

Q6. In Figure S1 panel A, there were two bands on PHRF1’s blot. Please indicate which band is PHRF1 and which is unspecific band.

A6. Figure S1 is moved to Figure 1D; therefore, we have labeled “nonspecific band” in Fig 1D. 

Q7. The complementary experiment in Figure S1 was a very solid evidence to confirm that PHRF1 KO directly compromised CSR progression. It is highly recommended to move these results to Figure 1.

A7. Many Thanks for this suggestion. We have moved Figure S1 to Figure 1D

Q8. The abbreviation of “CIT” should be described in the legend of Figure 1B rather than Figure 1C.

A8. We have added “CIT” in the figure legend of Figure 1B.

Q9. Lack of Data Availability statement.

A9. Data Availability statement is included in the cover letter.

Reviewer #2: 

Q1. The logic behind investigating transcriptional elongation in PHRF1 KO cells is not clearly defined. Even though, the PHRF1-deficient CH12 cells showed decreased phosphorylation of Rbp1 subunit of RNA polII at S2 position, the germline transcription at IgH locus in CH12 cells was unaffected. In addition, the effects of PHRF1 deficiency on NELF expression and Rbp1 phosphorylation were totally missing in activated primary B cells. The authors should also examine IgH germline transcription in PHRF1 KO primary B cells.

A1. Thanks for the advice. At first we were also confused with this conflict between the decreased pS2 of Rpb1 and unchanged GLTs. Our explanation is that the effect of PHRF1 KO on Rpb1 phosphorylation might “delay” the progression of transcription elongation, but not entirely abolish the elongation process. Therefore, GLTs were not changed in control and PHRF1 KO cells. Germline transcripts of I� and I� in PHRF1 KO primary B cells were shown in S8 Fig. There is no significant difference between control and PHRF1 KO B cells.

Q2. Consistent with the role PHRF1 in regulating TGFB signaling, there appears to be a slight increase in the levels of phospho-SMAD2/3 in PHRF1 KO CH12 cells, though this is not quantified. Can the authors test the possibility that the PHRF1-deficent CH12 and primary B cells are differentially sensitive to TGFB stimulation? One can do this by stimulating B cells in varying concentrations of TGFB and measuring the survival, proliferation and CSR to IgA in CH12 and primary B cells.

A2. Thanks for this suggestion. We have used different concentrations of TGF-� to treat control and PHRF1 KO primary B cells to address whether TGF-� was a key factor for IgA switching. In S7 Fig, there is no significant change between control and PHRF1 KO B cells in the proliferation using CFSE staining and CSR to IgA under the condition of different concentrations of TGF-�.

Q3. The authors should also provide a more thorough analysis of the transcriptomics data, especially with regards to the transcriptional factors that were identified as being differentially expressed. Perhaps a more careful examination of TGFB signaling could provide readers some more context.

A3. We have completed 14 genes’ expression using RT-qPCR and their results were shown in Fig 4B. Most of these genes were significantly down-regulated, which was in line with the RNA-seq results. To further address whether PHRF1 deficiency affected TGF-� signaling, we clustered two heatmaps containing entire components or fold change >2 in TGF-� signaling (KEGG#04350) in S4 Fig. The P-value of TGF-� signaling was 0.56 in the pathway analysis, suggesting that PHRF1 KO might not affect TGF-� signaling. 

Q4. The authors observed a slight but non-significant change in the microhomology usage. At what time point were these experiments performed. The repair of switch junctions through MMEJ follows much slower kinetics (PMID: 31806351) and therefore it is better to analyze the microhomology usage at 72 hours post-CIT stimulation. More clones should be screened to calculate the statistical significance of these findings.

A4. We isolated DNA from control and PHRF1 KO cells at 72 h post CIT treatment. We collected more clones (control n = 41; PHRF1 KO n = 37) to analyze MH. The results shown in Fig 2 and S4 Fig suggested that PHRF1 KO did not affect the length of microhomology.

Q5. The authors should refine the writing at several places in the manuscript.

A5. Thanks for the reminder. We invited Ms. Adaobi Amanna to improve our writing in this revised manuscript.

Q6. The statistical methods used in the study are not detailed enough. The authors should clearly describe the statistical methods used with each figure legend.

A6. We used the student’s t test and add it to each figure legend.

---

## [Decision Letter · Decision Letter 1]

17 Apr 2023

PHRF1 promotes the class switch recombination of IgA in CH12F3-2A cells

PONE-D-23-01559R1

Dear Dr. Chang,

We’re pleased to inform you that your manuscript has been judged scientifically suitable for publication and will be formally accepted for publication once it meets all outstanding technical requirements.

Kind regards,

Yan Shan, PhD

Academic Editor

PLOS ONE

Additional Editor Comments (optional):

Reviewers' comments:

Reviewer's Responses to Questions

**Comments to the Author**

1. If the authors have adequately addressed your comments raised in a previous round of review and you feel that this manuscript is now acceptable for publication, you may indicate that here to bypass the “Comments to the Author” section, enter your conflict of interest statement in the “Confidential to Editor” section, and submit your "Accept" recommendation.

Reviewer #1: (No Response)

Reviewer #2: All comments have been addressed

2. Is the manuscript technically sound, and do the data support the conclusions?

Reviewer #1: (No Response)

Reviewer #2: Partly

3. Has the statistical analysis been performed appropriately and rigorously? 

Reviewer #1: (No Response)

Reviewer #2: Yes

4. Have the authors made all data underlying the findings in their manuscript fully available?

Reviewer #1: (No Response)

Reviewer #2: Yes

5. Is the manuscript presented in an intelligible fashion and written in standard English?

Reviewer #1: (No Response)

Reviewer #2: Yes

6. Review Comments to the Author

Reviewer #1: (No Response)

Reviewer #2: The authors have at least attempted to address some of my suggestions, but the molecular phenotypes still appear to be a bit obscure. I don't think the authors can really do much in a reasonable time frame to provide insights into these phenotypes, so I suggest accepting the manuscript in its present form.

7. PLOS authors have the option to publish the peer review history of their article (what does this mean?). If published, this will include your full peer review and any attached files.

Reviewer #1: No

Reviewer #2: No

---

## [Editor Report · Acceptance letter]

25 Apr 2023

PONE-D-23-01559R1 

PHRF1 promotes the class switch recombination of IgA in CH12F3-2A cells 

Dear Dr. Chang:

I'm pleased to inform you that your manuscript has been deemed suitable for publication in PLOS ONE. Congratulations! Your manuscript is now with our production department. 

Kind regards, 

on behalf of

Dr. Yan Shan 

Academic Editor

PLOS ONE